# Effects of Multiple Environmental Stressors on Zoobenthos Communities in Shallow Lakes: Evidence from a Mesocosm Experiment

**DOI:** 10.3390/ani13233722

**Published:** 2023-12-01

**Authors:** Xiaoqi Xu, Guohuan Su, Peiyu Zhang, Tao Wang, Kangshun Zhao, Huan Zhang, Jinhe Huang, Hongxia Wang, Xianghong Kong, Jun Xu, Min Zhang

**Affiliations:** 1Institute of Hydrobiology, Chinese Academy of Sciences, Wuhan 430072, China; xuxiaoqi@ihb.ac.cn (X.X.); suguohuan@ihb.ac.cn (G.S.); zhangpeiyu@ihb.ac.cn (P.Z.); wangtao02@ihb.ac.cn (T.W.); zhaokangshun@ihb.ac.cn (K.Z.); zhanghuan@ihb.ac.cn (H.Z.); hongxiawang@ihb.ac.cn (H.W.); kongxh@ihb.ac.cn (X.K.); xujun@ihb.ac.cn (J.X.); 2College of Fisheries and Life Science, Dalian Ocean University, Dalian 116023, China; 3Hubei Provincial Engineering Laboratory for Pond Aquaculture, Engineering Research Center of Green Development for Conventional Aquatic Biological Industry in the Yangtze River Economic Belt, College of Fisheries, Huazhong Agricultural University, Wuhan 430070, China; huangjinhe@webmail.hzau.edu.cn

**Keywords:** climate change, eutrophication, pesticide contamination, multiple stressors, zoobenthos, α-diversity, β-diversity

## Abstract

**Simple Summary:**

This investigation delves into the intricate world of aquatic life, specifically zoobenthos, and how they respond to a combination of environmental challenges: climate change, eutrophication, and pesticide contamination. These organisms play a vital role in aquatic ecosystems, affecting energy flow, nutrient cycling, and sediment decomposition. Unfortunately, these challenges have led to a decline in their numbers and changes in community structure. Notably, the warmer temperatures associated with climate change promote the abundance and diversity of zoobenthos while making these communities more similar across different sites. Pesticides like imidacloprid negatively impact the survival and growth of zoobenthos. Interestingly, when combined with warming, imidacloprid seems to mitigate its adverse effects, increasing species diversity. However, when nutrient loading is part of the equation, imidacloprid negatively affects species diversity. These findings reveal the intricate responses of zoobenthos to multiple stressors, offering valuable insights for ecosystem conservation and management. In simpler terms, this study explores how tiny underwater creatures react to a changing environment, which has consequences for our ecosystems.

**Abstract:**

Multiple stressors, including climate change, eutrophication, and pesticide contamination, are significant drivers of the decline in lake zoobenthos. Zoobenthos play a crucial role in aquatic ecosystems, impacting energy dynamics, nutrient cycling, and sediment degradation. However, these stressors have led to a decrease in the abundance and diversity of zoobenthos, resulting in notable changes in species composition and structure. Eutrophication typically increases zoobenthos abundance while reducing taxonomic diversity. Climate change, such as warming and heatwaves, also affects the zoobenthos community structure, with different species exhibiting varying levels of adaptability to temperature changes. Additionally, pesticides like imidacloprid have negative effects on the survival and growth of zoobenthos. However, the interactions between imidacloprid and other stressors remain understudied. Here, we used 48 mesocosms (2500 L) to simulate shallow lakes. We combined nutrient loading, sustained warming, and the imidacloprid pesticide to test how these stressors interactively influence the survival and community of zoobenthos. The experimental results demonstrate that elevated temperatures have a significant impact on aquatic benthic organisms under different treatment conditions. The increase in temperature led to a notable rise in species richness and α-diversity, primarily attributed to the stimulation of metabolic activities in zoobenthos, promoting their growth and reproduction. This finding underscores the potential influence of climate change on aquatic benthic ecosystems, particularly in terms of its promoting effect on α-diversity. However, it is essential to note that elevated temperatures also reduced β-diversity among different sites, implying a potential trend toward homogenization in zoobenthos communities under warmer conditions. Moreover, this study revealed the interactive effects of multiple stressors on the diversity of aquatic benthic communities. Specifically, the pesticide imidacloprid’s impact on zoobenthos is not isolated but demonstrates complex effects within various treatment interactions. In the presence of both temperature elevation and the addition of imidacloprid, the presence of imidacloprid appears to counteract the adverse effects of temperature elevation, resulting in increased species diversity. However, when imidacloprid coincides with nutrient input, it significantly affects α-diversity negatively. These findings highlight the complexity of zoobenthos responses to multiple stressors and how these factors influence both α-diversity and β-diversity. They provide valuable insights for further research on the conservation and management of ecosystems.

## 1. Introduction

Zoobenthos organisms constitute pivotal constituents of shallow-water ecosystems, assuming an intermediary trophic position within aquatic food webs [1,2]. They exert a substantial influence on energy dynamics, nutrient cycling, and sediment degradation within river and lake aquatic systems [3,4,5]. Zoobenthos possess crucial regulatory functions and play a decisive role in maintaining and restoring the stability and integrity of river and lake ecosystems [6,7]. However, global environmental changes, including eutrophication, warming, and pesticide pollution, have caused a decline in zoobenthos abundance and diversity worldwide, leading to significant changes in species composition and structure [8,9,10].

Additive diversity partitioning is a promising method for understanding beta diversity patterns across different spatial and temporal scales and for analyzing diversity in hierarchical studies with multiscale sampling [11,12,13]. Additive diversity partitioning was proposed in the 1960s [14] and was revived following a theoretical analysis by Lande (1996). Beta diversity was originally calculated using a multiplicative relationship, where beta diversity (between-habitat diversity) was defined as the quotient of gamma diversity (total diversity of a landscape) divided by alpha diversity (within-habitat diversity) (i.e., β = γ/α) [15,16]. In the latter relationship, beta diversity (between-community diversity) is a dimensionless quantity. This implies that beta diversity cannot be given equal weight as alpha or gamma diversity, nor can it be directly compared across different components of diversity [11]. However, with additive diversity partitioning, beta diversity is calculated using an additive relationship (i.e., β = γ − α) [11]. In additive diversity partitioning, all diversity components are measured using the same method and expressed in the same units, enabling direct comparisons between them [17,18]. This approach allows for the quantification of the contributions of alpha and beta diversity to the overall diversity across various spatial and temporal scales [11,19]. Therefore, the utilization of additive diversity partitioning can readily identify the key sources of diversity in hierarchical studies with multiscale sampling [11]. However, this technology has not yet been used to study changes in the zoobenthos community structure in shallow lake ecosystems under multiple environmental stressors.

Excessive loading of nitrogen and phosphorus leads to increased eutrophication of water bodies, resulting in a higher abundance of zoobenthos but reduced taxonomic diversity [20]. Eutrophication tends to alter the habitat structure, directly leading to the extinction of sensitive zoobenthos species that cannot tolerate new abiotic conditions, thus reducing taxonomic differences in zoobenthos communities [21,22,23,24]. Additionally, anthropogenic eutrophication acts as an “ecological filter” by reducing the importance of stochastic processes in community structures, thereby reducing the compositional differences between different locations [25,26,27]. The underlying mechanism behind this process can be explained by “niche selection”, where strong environmental filtering excludes species that are less adapted to high-nutrient environments, and these species may undergo changes independent of habitat heterogeneity [28,29].

Climate change, including rising average water temperatures and heatwaves, also affects the community structure of zoobenthos organisms [24,30,31]. Various species within the zoobenthos community exhibit distinct levels of adaptability to diverse temperature conditions [24]. Climate warming will notably reshape the life history, biomass, density, and dimensions of zoobenthos toward smaller sizes [32,33,34]. Warming temperatures can advance the breeding time for species that are triggered by high temperatures, while shortening the breeding time for species triggered by low temperatures [35]. For instance, Chironomids, which favor breeding in warm water, may benefit from elevated temperatures as their growth and development rates accelerate with temperature, resulting in an increase in their abundance and alterations in the community structure of zoobenthos [36,37,38]. Moreover, climate change can indirectly impact zoobenthos by influencing the community structure of other biological groups such as algae and aquatic plants [39,40,41]. While the effects of climate change on density in lake ecosystems have been widely studied for various components like fish, phytoplankton, and zooplankton, the impact of climate warming on zoobenthos has been less explored.

Imidacloprid [1-(6-chloro-3-pyridylmethyl)-N-nitroimidazolidin-2-ylideneamine] is a novel neonicotinoid synthetic insecticide that is widely used for crop protection and pest prevention due to its broad-spectrum and high insecticidal activity [42,43,44]. One major advantage of imidacloprid is its selective action on the central nervous system of insects (post-synaptic nicotinic acetylcholine receptors) [45,46,47]. However, some studies have indicated that the use of neonicotinoid insecticides, such as imidacloprid, is a key factor in the decline in non-target terrestrial insect biodiversity in the environment [48,49,50,51,52]. Imidacloprid is highly soluble in water, and it is likely to enter aquatic ecosystems via runoff, posing a potential risk to non-target aquatic organisms [48,53]. It is already established that the aquatic life stages of insects are highly sensitive to imidacloprid exposure, and these toxic effects can have implications throughout the entire aquatic community [54,55,56]. As such, imidacloprid is known to exert deleterious effects on the survival and growth of zoobenthic organisms [57,58]. However, the effects of imidacloprid on zoobenthos, particularly in relation to interactions with other stressors, have received limited attention. Additionally, it is important to recognize that different zoobenthic species may show varying responses to multiple stressors [59,60].

In this study, we established a simulated shallow lake ecosystem employing 48 mesocosms and administered a combination of nutrient loading, warming, and imidacloprid to investigate their interactive effects on zoobenthos. Our principal hypotheses postulate that the cumulative influences of these concurrent environmental stressors will exert substantial, and potentially synergistic, effects on zoobenthos populations, with particular sensitivity exhibited by species acknowledged to be responsive to these stressors. We further hypothesize that the magnitude and character of these effects will diverge across distinct treatment conditions, reflecting the intricate interplay between stressor typologies and their respective intensities. Our expectations encompass discernible modifications in zoobenthos abundance, community composition, and diversity, with the prospect of disproportionate impacts on sensitive species within specified treatment contexts. This research endeavors to enhance our comprehension of the intricate ecological reactions of zoobenthos to the multifarious challenges presented by multiple environmental stressors, elucidating the variable responses contingent upon differing treatment scenarios.

## 2. Materials and Methods

### 2.1. Experiment Design

The experiment was conducted at Huazhong Agricultural University, Wuhan, China (30°29′ N; 114°22′ E), from February to November 2021. We employed 48 cylindrical polyethylene mesocosms (with a volume of 2500 L, a diameter of 1.5 m, and a height of 1.4 m) that were buried up to their rims. This was chosen to enhance insulation against prevailing weather conditions and to replicate shallow lake ecosystems. In the experiment, we applied three stressors as treatments: warming (W), nutrient loading (E), and the introduction of imidacloprid (P). These three stressors were randomly assigned to all mesocosms, including controls without stressors, in a fully factorial design. This resulted in eight treatments, each with six replicates (Figure 1).

The warming treatment entailed maintaining temperatures at a constant +3.5 °C above the ambient conditions throughout the experiment, in addition to the introduction of multiple heatwave events, and the frequency and magnitude of the heat waves was based on model predictions from the historical meteorological data in the middle and lower reaches of the Yangtze River Basin, China, predicted to be reached in this area by the end of this century given the ongoing climate warming (IPCC 2014). The warming was achieved using a computer-controlled system with two temperature sensors (DS18B20) and a heating element (600 W) in each of the heated treatment mesocosms. The heating element was installed 30 cm below the water surface and an aquarium pump was closely placed to circulate the water. The water temperatures in the heated mesocosms were elevated based on the mean temperatures in the ambient mesocosms [61,62,63,64]. Nitrogen (N) and phosphorus (P) were added to the nutrient-loading treatment at a mass ratio of 10:1, by dissolving NaNO_3_ and KH_2_PO_4_ powder (Sinopharm Chemical Reagent Co., Ltd., Shanghai, China) in de-mineralized water, respectively. The averaged nutrient-loading doses were 0.90 mg L^−1^ (range from 0.25 to 1.6 mg L^−1^) and 0.09 mg L^−1^ (range from 0.025 to 0.16 mg L^−1^) for N and P, respectively. Insecticide treatments were applied by adding imidacloprid (70% active ingredients, PD20120072, Bayer, Leverkusen, Germany) solution to the mesocosms. The average insecticide-loading dose was 32.67 µg L^−1^ (range from 10 to 50 µg L^−1^) during the experiment. The nutrient-loading and insecticide treatments were administered every two weeks, with dose adjustments made in response to agricultural activities and precipitation levels in the area [65]. This approach aimed to replicate more realistic scenarios involving temporally varying multiple stressors [66]. The loading doses were within the concentration range typically observed in natural water bodies in agricultural areas around the world [67,68].

### 2.2. Experiment Set-Up

The experiment was set up in early February of 2021, as it was winter, when organisms were in dormancy. The mesocosms were acclimated for two months and all treatments were initiated on April 8, 2021. Half of the bottom of each mesocosm was filled with 10 cm of sediment, which was collected from Lake Liangzi (30°11′3″ N, 114°37′59″ E). *Potamogeton crispus* and *Hydrilla verticillata* are the dominated submerged macrophytes in this area. All sediment was homogenized and sieved through a 5 × 5 mm metal mesh to remove large debris, macrophyte seeds, and snails. The initial properties of the sediment were a total nitrogen (TN) of 5.5 ± 0.4 mg g^−1^ and total phosphorus (TP) of 0.42 ± 0.08 mg g^−1^, dry weight. Turions of *Potamogeton crispus* and *Hydrilla verticillata* were seeded in the sediment, each species comprising 50 g in each mesocosm. The turions were obtained from the nearby Lake Honghu (29°51′ N, 113°20′ E). *Potamogeton crispus* is an early season submerged macrophyte dominating in spring, while *Hydrilla verticillata* is a warm-adapted species dominating in summer.

The water level was gradually raised using tap water and rainfall to a 1.2 m depth to allow the establishment of the submerged macrophytes. Before the experiment began in April, our goal was to simulate a natural food web in the mesocosms by introducing 14 individuals of *Radix swinhoei* (1 to 2.5 cm) and 20 individuals of *Bellamya aeruginosa* (around 2.5 cm) into each mesocosm as periphyton grazers. Five freshwater shrimps *Macrobrachium nipponense* (length around 4 cm), four bitterling *Rhodeus sinensis* (around 3 cm), and four crucian carp *Carassius auratus* (around 4 cm) were added as omnivores feeding on zooplankton, macroinvertebrates, detritus, periphyton, and phytoplankton. The fish and shrimp were commercially obtained, but are common species coexisting in water bodies in this region, and the densities and biomasses were within the range occurring in nature [2,69,70,71]. Also, 10 L of lake water were added to each mesocosm to inoculate plankton from the nearby Lake Nanhu as a common garden inoculum (30°28′57″ N, 114°22′34″ E). An aquarium pump was installed in each mesocosm to allow for mixing of the water. Deionized water was added to the heated mesocosms to compensate for evaporation. The submerged macrophyte *Ceratophyllum demersum* and floating macrophyte *Lemna minor* emerged in a few mesocosms and were removed as soon as they were observed. Dead fish were recorded and removed during the experiment (Figure 2).

### 2.3. Sampling Strategy

All treatments were applied when the macrophytes had established and the water was clear in all mesocosms on 8 April 2021. Water quality samples were measured bi-weekly, including for water conductivity, pH, turbidity, TN, TP, and phytoplankton chlorophyll a. The conductivity and pH were measured using HACH HQD Portable Meters (HQ60d, HACH, Loveland, CO, USA). The turbidity was measured using a portable WGZ-2B turbidity meter (Xinrui, Shanghai, China). Depth-integrated water samples were collected using a transparent Plexiglas tube (diameter 70 mm, length 1 m) to analyze the TN, TP, and phytoplankton chlorophyll a concentration. The TN and TP were first digested using potassium peroxodisulfate, and then measured using the spectrophotometric method (Chinese National Standards [72]). The phytoplankton chlorophyll a was determined by filtering a certain amount of water through Whatman GF/C filters and conducting spectrophotometric analysis after acetone extraction (HJ 897-2017) (Chinese National Standards [72]).

Zoobenthos: After the start of the experiment, zoobenthos were collected approximately every 30 days. Larger gastropods, such as the snail species *Radix swinhoei* and *Bellamya aeruginosa*, were initially collected from the walls and sediment. To ensure that all large individuals were accounted for, a second search was conducted after leaving the mesocosm overnight. Other zoobenthos were quantified using a macroinvertebrate collection metal basket filled with pebbles ranging from 1 to 6 cm in size. This basket was placed on the sediment and then removed, and all pebbles were thoroughly rinsed to collect all zoobenthos, following the protocol used by Brock et al. [73]. During the experiment, zoobenthos were collected once each month in April, May, June, July, August, October, and November. All species were identified to the greatest extent possible in terms of taxonomy and subsequently grouped into three taxon categories, *Insecta*, *Oligochaeta*, and other small snails, for the analysis (Figure 2).

### 2.4. Data Analysis

All statistical analyses were performed using R software version 4.2.2 (R Core Team, 2022) (R code data in Appendix A). At the conclusion of the experiment, a Poisson distributed generalized linear model was fitted using the “glm” function from the “stats” package to compare the abundance of zoobenthos in different treatment groups with that of the control group. In this model, the abundance of zoobenthos in various treatment groups served as the response variable, while temperature, eutrophication, and pesticide exposure were considered the predictor variables. It was verified that the model residuals conformed to a normal distribution. Subsequently, the “emmeans” function from the “emmeans” package [74] was applied to the fitted model. Pairwise comparisons among the levels of the interaction term were calculated using the “pairwise” parameter, and multiple comparisons were adjusted using Tukey’s correction with the “adjust” parameter set to “tukey”. These tests were performed to assess specific differences in zoobenthos abundance between different combinations of warming, eutrophication, and pesticide exposure levels, while controlling for the overall interaction effects.

In order to delve into the potential impacts of warming, eutrophication, and pesticides on zoobenthos diversity (bottom-dwelling aquatic organisms), we conducted additive diversity partitioning analysis. We employed the “adipart” function from the “vegan” package [75] to calculate the α, β, and γ components of zoobenthos richness, the Shannon diversity index, and the Simpson diversity index, taking into account the sample abundance proportions. The Shannon diversity index measures the overall diversity of a community by considering both the abundance and evenness of different species, while the Simpson diversity index focuses more on the dominance of a few highly abundant species. The key difference is that Shannon is more sensitive to species evenness, whereas Simpson emphasizes species dominance [76]. These computations were performed using 999 permutation simulations to estimate the statistical significance of the zoobenthos diversity components. Finally, the “quantile” function was employed to determine the lower, median, and upper limits of different diversity indices for the α, β, and γ components of zoobenthos diversity. 

To analyze the differences in the biodiversity indices and precisely quantify these differences, we employed the “effsize” package [77] for comprehensive effect size analysis. Specifically, we utilized the “cohen.d” function to calculate Cohen’s d effect sizes between the treatment and control groups at the α and β levels. These effect size metrics encompassed estimates, confidence intervals, and magnitude assessments. This meticulous approach aimed to evaluate the significance of the observed differences in biodiversity indices between the two groups at the α and β levels and elucidate the impact of various treatments on the biodiversity indices [78]. “Cohen’s d” functioned as a robust metric for measuring the effect size of the mean differences between the treatment and control groups, effectively accounting for the data variability within each group [78]. Its calculation method is based on the following formula:d = (M_1_ − M_2_)/s

Here, d represents the Cohen’s d effect size, M_1_ signifies the mean of the biodiversity indices under varying treatment conditions, M_2_ denotes the mean of the biodiversity indices within the control group, and s corresponds to the standard deviation of the aggregated biodiversity indices. When Cohen’s d crosses the zero mark, it indicates that there is no statistically significant effect. If Cohen’s d < 0, it signifies a statistically significant negative effect of the treatment on diversity. If Cohen’s d > 0, it signifies a statistically significant positive effect of the treatment on diversity.

To analyze the response of different benthic animal species to various treatment factors, a Gaussian-distributed linear mixed-effects model was fitted using the “lmer” function from the “lmerTest” package. In this model, the logarithm of the abundance of zoobenthos for different species served as the response variable, while temperature, eutrophication, and pesticide exposure were treated as fixed-effect variables. Conducting the Shapiro–Wilk normality test using the “shapiro.test” function resulted in a *p*-value greater than 0.05, indicating conformity with a normal distribution. Additionally, the month was included as a random variable. Subsequently, the significance levels (*p*-values) and the direction of effects (positive or negative) from the computed results were stored in Table 1.

## 3. Results

### 3.1. Warming Effects on Zoobenthos Diversity

At the conclusion of the experiment, the warming group (W) exhibited a significant increase in the abundance of zoobenthos compared to the control group (C) (Figure 3). During the experiment, regarding biodiversity, under the warming treatment, the zoobenthos richness significantly increased at both the α and β levels. The Shannon index showed a significant increase at the α level while exhibiting a significant decrease at the β level. Similarly, the Simpson index demonstrated a significant increase at the α level but a significant decrease at the β level (Figure 4).

### 3.2. Eutrophication Effects on Zoobenthos Diversity

At the conclusion of the experiment, the abundance of zoobenthos in the eutrophication group (E) did not show a significant change compared to the control group (C) (Figure 3). In terms of biodiversity, nutrient loading increased the richness at the α level but decreased it at the β level. Both the Shannon and Simpson indices decreased significantly at both the α and β levels (Figure 5).

### 3.3. Pesticide Effects on Zoobenthos Diversity

At the conclusion of the experiment, the imidacloprid addition group (P) significantly reduced the abundance of zoobenthos compared to the control group (C) (Figure 3). During the experiment, in terms of biodiversity, under the pesticide treatment, the richness significantly increased at the α level but decreased at the β level. The Shannon index increased significantly at both the α and β levels, as did the Simpson index (Figure 6).

### 3.4. All Treatments’ Effect on Zoobenthos Diversity and the Response of Zoobenthos Abundance to the Treatments

At the end of the experiment (with C as the control group), warming (W), nutrient loading (E), pesticide application (P), and their interactions had varying impacts on zoobenthos. The highest zoobenthos abundance was under the warming and nutrient loading (WE) treatment, while the lowest was under nutrient loading and pesticide application (EP) (Figure 3). In terms of biodiversity, the E treatment reduced the richness at the α level, whereas the EP, P, W, WE, WEP, and WP treatments increased the richness at the α level. All treatments showed a decrease at the β level. For the Shannon index, the E and EP treatments decreased at the α level, while the P, W, WE, WEP, and WP treatments increased at the α level. The E, EP, WE, WEP, and WP treatments decreased at the β level, while the P and W treatments increased at the β level. Regarding the Simpson index, the E, P, W, WEP, and WP treatments increased at the α level, whereas the EP and WE treatments decreased. The E, EP, WE, WEP, and WP treatments decreased at the β level, with the P treatment increasing, and the W treatment showing no significant change at the β level (Figure 7).

There were different zoobenthos abundance responses for different treatments. All treatments had significant effects on some of the top 35 species in abundance and none on the rest, and each treatment had a different direction of positive or negative effects on each species (Table 1).

## 4. Discussion

### 4.1. The Effects of Different Treatment Conditions on the α Diversity of Zoobenthos

Our findings align with previous research, highlighting that increasing temperatures positively influence zoobenthos species abundance and α-diversity [79,80]. Elevated temperatures stimulate metabolic activities, fostering a more diverse community. Favorable conditions, driven by temperature, may attract a broader range of species [81,82]. In contrast, nutrient loading has a mixed impact on α-diversity. While it promotes species richness, it negatively affects the Shannon and Simpson diversity indices [83]. Recent research emphasizes ecological networks’ role in these dynamics [84]. Nutrient loading fosters eutrophication, favoring dominant species and disrupting ecological interactions. This shift reduces the species distribution evenness by increasing the dominance of certain species. Imidacloprid addition positively influences zoobenthos α-diversity, indicating that imidacloprid may selectively target specific organisms, reducing competition and benefiting other species [48,85]. However, further research is needed to understand the underlying mechanisms and long-term consequences.

The WE treatment, which combined warming and nutrient loading, exhibited an increase in species richness and Shannon diversity but a decrease in Simpson diversity. This result suggests that the combination of these stressors may have created favorable conditions for a wider range of species to coexist [86]. Warming might have accelerated metabolic rates, while nutrient loading provided additional resources, promoting diversity [87]. In the EP treatment group (nutrient loading and imidacloprid), the increase in species richness combined with decreased Shannon and Simpson diversity may be attributed to the disruptive impact of imidacloprid [48,85]. While nutrient loading supported higher species richness, the presence of imidacloprid could have led to the dominance of certain pesticide-tolerant species, reducing the evenness. The WP treatment (warming and imidacloprid) resulted in increased species richness, Shannon diversity, and Simpson diversity. Warming may have alleviated some of the negative effects of imidacloprid, allowing for a more diverse community [88,89]. Elevated temperatures could enhance metabolic rates and promote greater resource utilization, counteracting imidacloprid’s potential disruptiveness [90]. The WEP treatment, which included all three stressors, exhibited increased species richness and diversity indices. This complex response may be attributed to the combined and possibly compensatory effects of these stressors [91,92]. Warming might enhance nutrient cycling, mitigating the negative effects of imidacloprid and supporting a more diverse community [93].

### 4.2. The Effects of Different Treatment Conditions on the β-Diversity of Zoobenthos

Our study, conducted in freshwater ecosystems, unveiled intriguing insights into the impacts of warming, nutrient loading, and the addition of the insecticide imidacloprid on zoobenthic community β-diversity. These findings emphasize the dynamic nature of benthic ecosystems and the multifaceted responses of their communities. Warming plays a significant role in reshaping the structure of zoobenthic communities at the regional scale (β-diversity) [94]. The decrease in β-diversity implies that warming exerts a homogenizing influence on the composition of zoobenthic communities across various sampling sites. Recent research has shed light on the potential mechanisms driving this phenomenon [80]. Warmer temperatures can enhance the metabolic rates of certain species (e.g., Chironomidae), giving them a competitive advantage and allowing them to dominate across multiple sites. Such dominance can lead to a reduction in regional variability, contributing to the observed decrease in β-diversity [95,96]. Conversely, nutrient loading exhibited a contrasting pattern in its impact on β-diversity. Despite reducing the species richness at both local and regional scales, it contributed to an increase in β-diversity, indicating greater differentiation in zoobenthic community composition among sites [97]. Eutrophication-driven nutrient enrichment can create favorable conditions for certain species, allowing them to thrive and dominate specific sites. These contrasting responses across sites contribute to the increased β-diversity in eutrophic environments [97]. Lastly, the introduction of the insecticide imidacloprid had a pronounced influence on zoobenthic community β-diversity. While causing a decrease in species richness at the local scale, it significantly increased β-diversity, signifying a greater variation in community composition among sites. Recent research has indicated that the selective toxicity of imidacloprid toward specific species can lead to shifts in community composition [98,99]. Furthermore, imidacloprid can indirectly affect the community structure by altering the abundance and distribution of primary producers (e.g., algae), which serve as the foundation of the food web for zoobenthic organisms [41,98,100]. These complex interactions result in the observed increase in β-diversity in imidacloprid-treated sites [101].

The consistent decrease in β-diversity across treatment groups (WE, EP, WP, and WEP) aligns with previous studies investigating the impacts of multiple stressors on aquatic ecosystems, reaffirming the robustness of our findings [102,103]. Several factors may contribute to the observed decline in β-diversity, indicative of a shift toward more similar community compositions among different sites or treatments, as observed in prior research. Our results are in line with previous research indicating that the decline in β-diversity may be attributed to the reduced presence or even disappearance of specialist species uniquely adapted to specific environmental conditions [104,105,106]. Consistent with prior studies, our findings suggest that warming, nutrient loading, and pesticide exposure can homogenize environmental conditions across sites, diminishing the variation supporting diverse community compositions [107,108]. Our results align with research indicating that the interactive effects of stressors could displace certain species that were previously dominant or unique to particular sites [109]. The reduction in β-diversity may also indicate functional redundancy within zoobenthic communities, a concept consistent with previous ecological research [110,111].

### 4.3. The Sensitivity of Different Zoobenthic Species to Various Treatment Conditions

*Ephemeroptera*, *Plecoptera*, and *Trichoptera* (EPT) are highly sensitive to environmental disturbances among large benthic organisms [112]. They are commonly used indicators for monitoring the health of freshwater ecosystems [112]. Based on the experimental results, five species of EPT were identified. Among them, the *Ecnomus* sp. ranked third in abundance among all benthic organisms, exerting a significant impact on the entire ecosystem. The four remaining species exhibited low abundances, each with fewer than 50 individuals, and can be regarded as negligible. The *Ecnomus* sp. typically inhabits cleaner, fast-flowing water bodies. Imidacloprid, as an insecticide, may accumulate in water, leading to potential water quality degradation in the habitats of the *Ecnomus* sp., affecting its survival and reproduction [113,114,115]. The larvae of the *Ecnomus* sp. primarily feed on aquatic microorganisms and algae [116]. The presence of imidacloprid may have adverse effects on the aquatic microbial communities, thereby impacting the food sources for the *Ecnomus* sp. larvae, potentially limiting their growth [117]. Imidacloprid has been demonstrated to inhibit reproduction and development in many insects [118,119,120]. If the *Ecnomus* sp. is subjected to imidacloprid inhibition, it may affect its reproduction and larval development, reducing the number of subsequent generations. Furthermore, prolonged exposure of the *Ecnomus* sp. to high concentrations of imidacloprid may result in more pronounced negative effects.

The *Chironomus* sp. is the most abundant species, dominating the entire ecosystem. Both warming and imidacloprid have a positive impact on the *Chironomus* sp., with imidacloprid having a stronger effect. Higher temperatures may enhance the *Chironomus* sp.’s activity and reproduction, while imidacloprid could improve resource utilization via nervous system or metabolic effects [57,121]. In warmer conditions, the *Chironomus* sp. may reproduce more easily, with faster larval development, shortening the generation time [122]. Imidacloprid might somehow boost reproduction and development via complex mechanisms [123]. Elevated temperatures usually promote the growth of algae and plankton, providing extra food. The *Chironomus* sp. may benefit from this, increasing its survival and reproductive success. Imidacloprid may alter the aquatic food chain, making food more accessible to the *Chironomus* sp. [124].

*Radix swinhoei*, as the most abundance *Mollusca*, plays an indispensable role in the ecosystem. Warming, nutrient loading, and imidacloprid all have a positive impact on *Radix swinhoei*, with warming and imidacloprid having a more pronounced effect. Elevated temperatures can enhance metabolism and growth, accelerating reproduction and population growth. Nutrient loading, representing increased nutrient levels, fosters the growth of algae and plants, which constitute *Radix swinhoei*‘s primary food source [125]. In nutrient-rich waters, they can access more food resources, facilitating their growth and reproduction [126]. Imidacloprid, despite being an insecticide, may positively influence *Radix swinhoei* by affecting food resources or ecological interactions under conditions of prolonged and continuous exposure [127,128,129]. Moreover, *Radix swinhoei* populations may gradually adapt to nutrient-rich conditions, warming temperatures, and imidacloprid exposure, potentially developing resistance to these factors over time.

### 4.4. Issues in the Study and Future Exploration

In our research, we identified significant variations in zoobenthos abundance between treatments, a limited range of studied species, and challenges quantifying snail biomass due to their larger size. To address these concerns, future studies can employ standardized methods to minimize the differences between treatments [130,131], expand the scope of investigated zoobenthos species, and employ modeling techniques for biomass estimation despite variations in species counts. Prioritizing functional assessments and establishing long-term monitoring programs will augment our comprehension of zoobenthic ecosystems and provide valuable insights for effective management strategies [132].

## 5. Conclusions

In conclusion, this study highlights the complex impacts of multiple stressors on the diversity of lake zoobenthos. Climate change and pesticides like imidacloprid play pivotal roles in zoobenthic communities, promoting α-diversity and reducing β-diversity. Subsequent research should delve into the ecological mechanisms underlying the responses of zoobenthos to multiple stressors and focus on effective management and conservation strategies for aquatic ecosystems to preserve their functionality and services.

## Figures and Tables

**Figure 1 animals-13-03722-f001:**
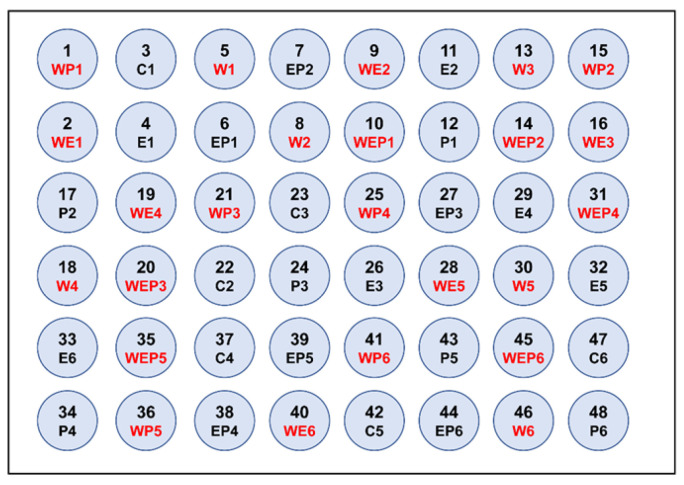
Distribution map of 48 mesocosms, including mesocosm numbers and types of experimental treatments.

**Figure 2 animals-13-03722-f002:**
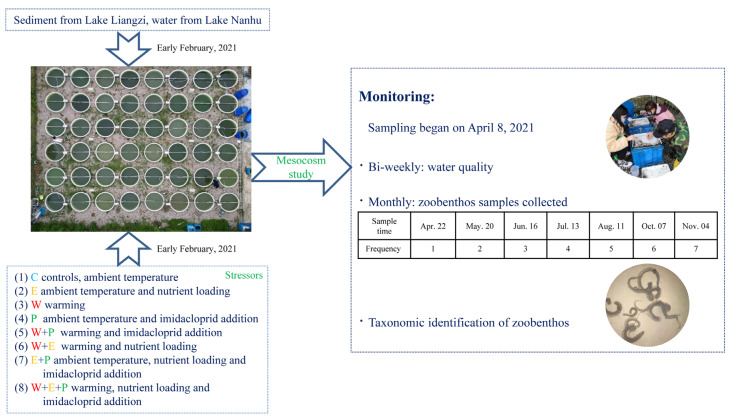
Schematic diagram of experimental process and time.

**Figure 3 animals-13-03722-f003:**
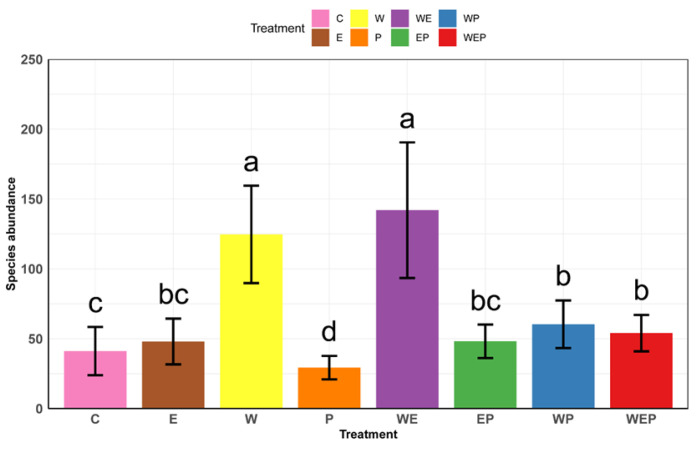
The effects of different treatments on zoobenthos abundance were examined at the end of the experiment. The treatments included warming (W), nutrient loading (E), and pesticide application (P), both individually and in combination. The ambient control group with no treatment added is denoted as (C). Lowercase letters represent significant differences in means between different treatments (post hoc tests, *p* < 0.05), while the same letters indicate no significant differences. Vertical bars are standard errors.

**Figure 4 animals-13-03722-f004:**
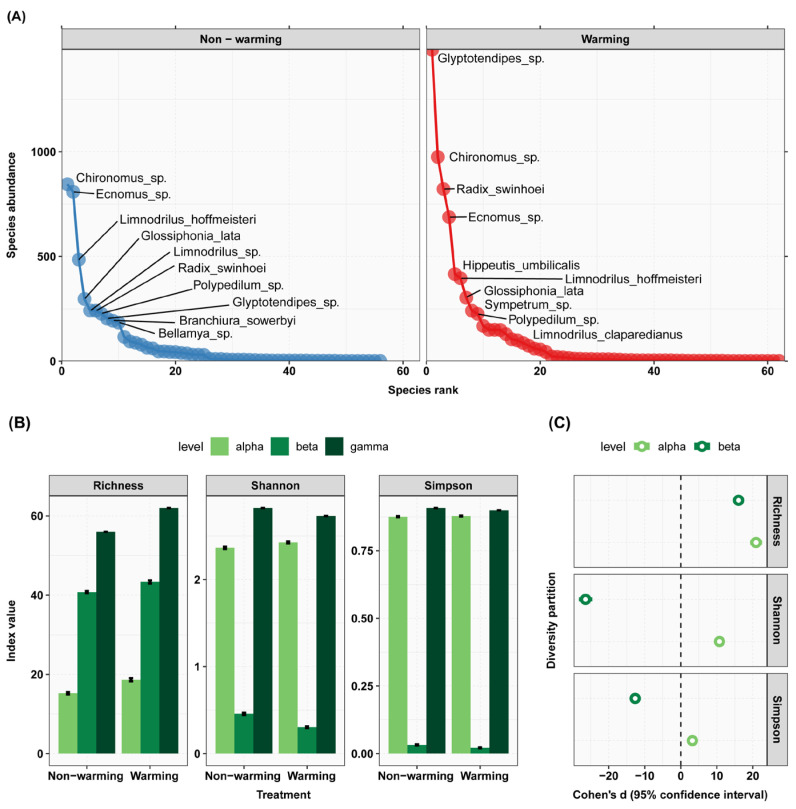
The impact of warming on zoobenthos species abundance and diversity. (**A**) Rank abundance curves: the figure illustrates species abundance ranking on the *x*-axis, with species abundance on the *y*-axis, and displays the names of the top 10 species as label text. (**B**) Additive diversity partitioning: the figure displays diversity indices on the *y*-axis for different treatment types on the *x*-axis, with separate graphs for richness, the Shannon index, and the Simpson index. (**C**) Cohen’s d: the *x*-axis of the figure represents the magnitude of Cohen’s d value, while the *y*-axis represents levels of diversity partitioning.

**Figure 5 animals-13-03722-f005:**
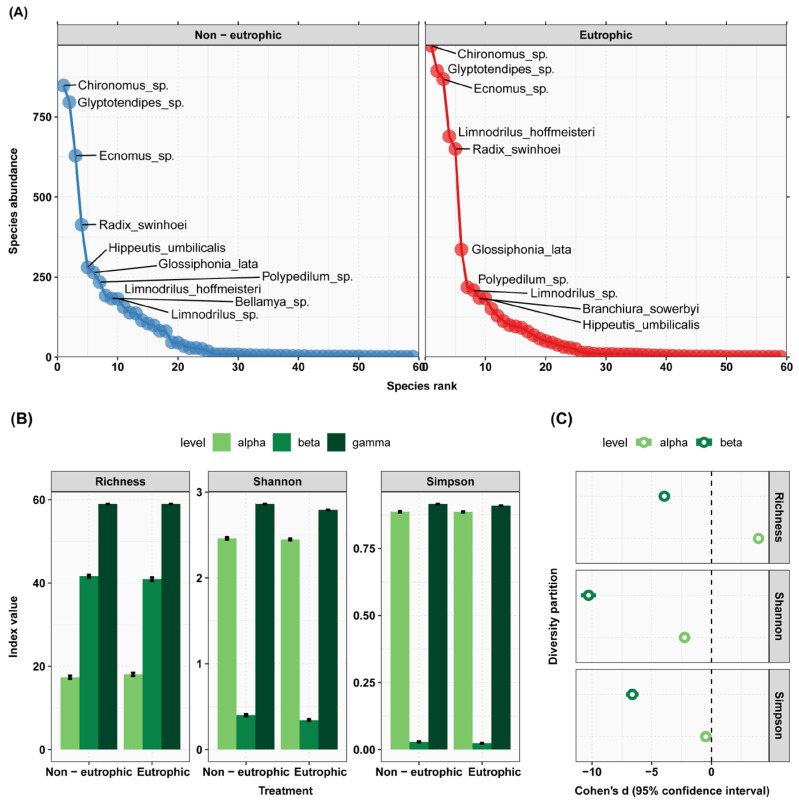
The impact of eutrophication on zoobenthos species abundance and diversity. (**A**) Rank abundance curves: the figure illustrates species abundance ranking on the *x*-axis, with species abundance on the *y*-axis, and displays the names of the top 10 species as label text. (**B**) Additive diversity partitioning: the figure displays diversity indices on the *y*-axis for different treatment types on the *x*-axis, with separate graphs for richness, the Shannon index, and Simpson index. (**C**) Cohen’s d: the *x*-axis of the figure represents the magnitude of Cohen’s d value, while the *y*-axis represents levels of diversity partitioning.

**Figure 6 animals-13-03722-f006:**
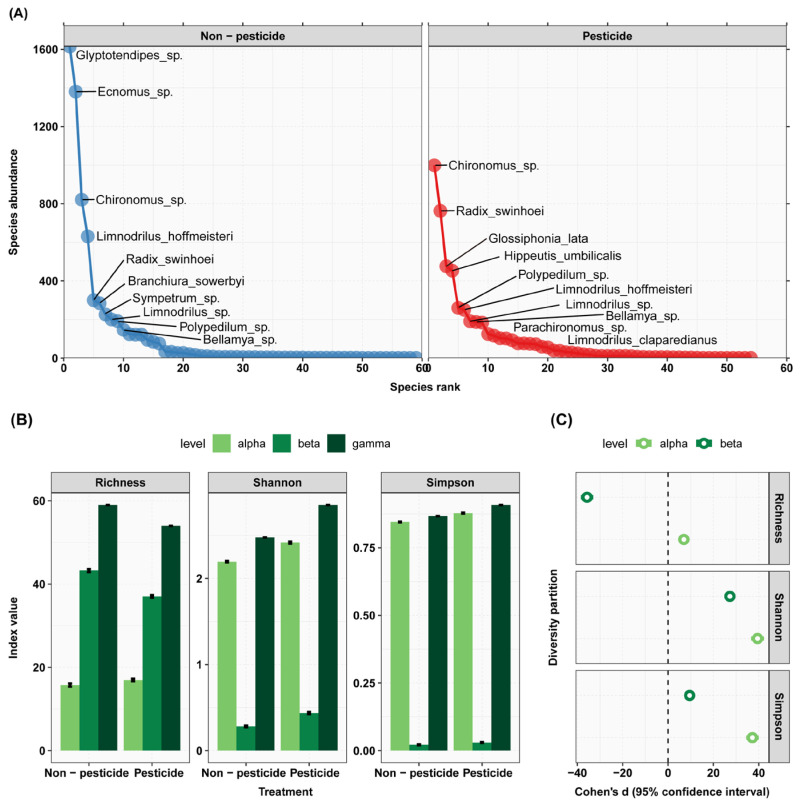
The impact of pesticide on zoobenthos species abundance and diversity. (**A**) Rank abundance curves: the figure illustrates species abundance ranking on the *x*-axis, with species abundance on the *y*-axis, and displays the names of the top 10 species as label text. (**B**) Additive diversity partitioning: the figure consists of three separate graphs representing “Richness”, “Shannon”, and “Simpson” indices. The *y*-axis represents diversity indices, while the *x*-axis represents different treatment types. (**C**) Cohen’s d: the figure consists of seven separate graphs, each representing different treatment type groups. The *x*-axis represents the magnitude of Cohen’s d value, while the *y*-axis represents levels of diversity partitioning.

**Figure 7 animals-13-03722-f007:**
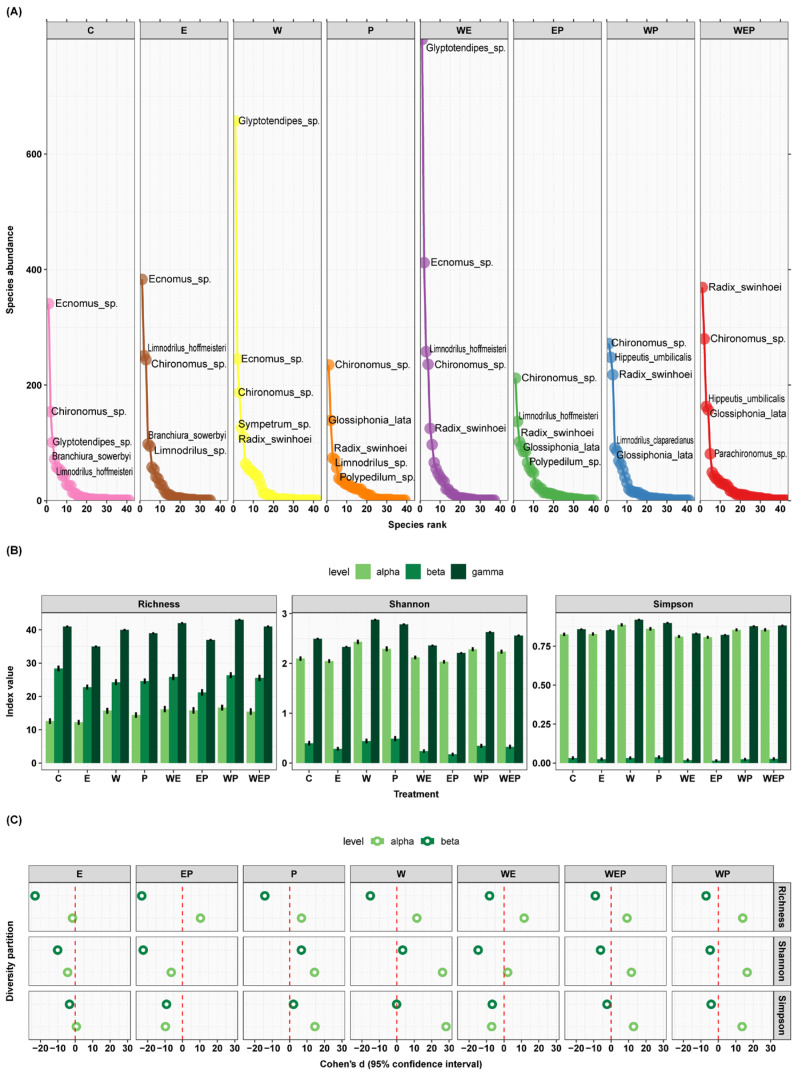
The impact of all treatments on zoobenthos species abundance and diversity. (**A**) Rank abundance curves: the figure illustrates species abundance ranking on the *x*-axis, with species abundance on the *y*-axis, and displays the names of the top 10 species as label text. The curves are color-coded to represent different treatment types. (**B**) Additive diversity partitioning: the figure displays diversity indices on the *y*-axis for different treatment types on the *x*-axis, with separate graphs for richness, Shannon index, and Simpson index. (**C**) Cohen’s d: the *x*-axis of the figure represents the magnitude of Cohen’s d value, while the *y*-axis represents levels of diversity partitioning.

**Table 1 animals-13-03722-t001:** Effects of temperature, eutrophication, pesticide, and their interactions on zoobenthos abundance at the end of the experiment. Accumfreq represents the cumulative proportion of species abundance. “***” represents highly significant (*p* ≤ 0.001), “**” represents significant (0.001 < *p* ≤ 0.01), “*” represents moderately significant (0.01 < *p* ≤ 0.05), “NS” represents not significant (*p* > 0.1); “(+)” represents positive correlation (r > 0) and “(−)” represents negative correlation (r ≤ 0). (Top 10 species by abundance in the table, complete table in Appendix A).

Species Name	Rank	Abundance	Accumfreq	Warming	Eutrophic	Pesticide
*Chironomus* sp.	1	1820	15.5	** (+)	NS	*** (+)
*Glyptotendipes* sp.	2	1690	29.9	NS	NS	* (−)
*Ecnomus* sp.	3	1497	42.6	NS	NS	*** (−)
*Radix swinhoei*	4	1063	51.7	*** (+)	* (+)	*** (+)
*Limnodrilus hoffmeisteri*	5	881	59.2	* (−)	** (+)	** (−)
*Glossiphonia lata*	6	600	64.3	NS	NS	* (+)
*Hippeutis umbilicalis*	7	463	68.2	NS	NS	* (+)
*Polypedilum* sp.	8	452	72.1	NS	NS	NS
*Limnodrilus* sp.	9	391	75.4	NS	NS	NS
*Bellamya* sp.	10	334	78.2	NS	NS	NS

## Data Availability

Data are contained within the article and Appendix A.

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
