# Peer review of "Effects of Multiple Environmental Stressors on Zoobenthos Communities in Shallow Lakes: Evidence from a Mesocosm Experiment"

_animals, 2023, doi:10.3390/ani13233722_

Round 1
Reviewer 1 Report
Comments and Suggestions for Authors
Minor changes.
Line 53: You are stating ‘In some instances’. Please be more specific.
Line 201: please delete extra space before comma.
Lines 234- 235: please state days of the month where zoobenthos were collected e.g. every 30 days ensuring no frequent disturbance of the population.
Line 279: please add reference.
Lines 309, 330, 351, : Please add number of warming vs non- warming treatments (n=?).
Line 430: please add reference (Elevated temperatures could enhance metabolic rates 429and promote greater resource utilization).
Line 433-434: please add reference (Warming might 433enhance nutrient cycling).
Line 516: please specify in which cases imidacloprid, may positively influence snails.
Reviewer 2 Report
Comments and Suggestions for Authors
The manuscript by Xu and colleagues entitled “Effects of multiple environmental stressors on zoobenthos communities in shallow lakes: evidence from an mesocosms experiment.” Describes the consequences of a warming climate, eutrophication, and pesticide exposure on zoobenthos diversity in a mesocosm experiment. This manuscript is extremely well written, clearly states hypothesis, provides a sound experimental design, and discusses the results and ramifications of the experimentation in clear detail. This is one of the best manuscripts I have reviewed in the past few years! The results are compelling and provide evidence of the complex interactions among multiple stressors in natural environments. In my opinion, this manuscript only needs the most minor edits (see below) to be publishable.
Minor Edits:
· The abbreviations used to describe the three stressors (warming W; eutrophication E; and pesticides P) are fairly intuitive. However, things get confusing as “P” is introduced in the context of phosphorus (line 92), “N” is introduced for nitrogen, and the pesticide imidacloprid is introduced as "I”. In Line 155, “N” is used for “nutrient loading” but in Table 1 W,P,E are used (but nor described in the legend to Table 1). In other words, the letters “N” and “P” are used for different things in different parts of the manuscript and figures/tables.
· A schematic timeline would be helpful to explain when the mesocosms were first setup, when they were seeded, when stressors where introduced, when sampling occurred, etc. There is a lot going on in this study and it would help readers visualize the chronology of events a bit better.
· Color accessibility (for people with deficiencies in being able to distinguish between colors) is generally very good in this manuscript (it helps that headings often identify the columns in addition to the colors). The only exception are the Cohen’s d value graphs in Figs. 3,5,7,9. Those circles could use greater contrast, or even a different shape to help distinguish them without referencing color.
· Table 1. This is a long manuscript (because it contains much information). I wonder if Table 1 could be moved to the supplemental information?
Reviewer 3 Report
Comments and Suggestions for Authors
Dear authors, your work deals with the influence of three stress factors on the zoobenthos community. This article addresses a very interesting topic and I believe it is necessary. I am now proceeding with my review:
The introduction seems appropriate as it introduces all the key points of the manuscript. Additionally, the references used are suitable.
135. It is not necessary to be so specific by indicating 2500 L, as it will be stated in the materials and methods section.
151-153. It would be convenient to specify how the mesocosms were positioned. You mention that they were buried, but adding some more detail about their placement would be advisable.
155. It is not necessary to indicate a common insecticide, as it has already been clarified previously that it is imidacloprid.
158. The meaning of the acronyms in Figure 1 should be clarified through a legend.
162. What were the environmental conditions in the controls?
174. Indicate the source of the powders.
178-179. This phrase is not necessary, "a common insecticide belonging to the neonicotinoids, used worldwide in agricultural areas."
197. Be consistent with the abbreviations of scientific names.
200-201. Spelling error.
212. Why was deionized water chosen? Justify this.
213. Use italics for "Lemma minor."
234. Cite as follows: "following the protocol used by Brock et al [72]."
239-293. I find the wording of the data analysis to be very well done. My question is the following: you used several parametric tests that imply meeting certain assumptions (normality or equality of variances, mainly). Was this ensured? It is true that you mention a normal (Gaussian) distribution, but it is not mentioned whether it was verified.
305-309. Perhaps the title of the figure is excessively long. It is not necessary to indicate what is shown on the axes as it is visible in the graph. The same applies to other figures.
311. Double space.
339. I don't think it is appropriate to use the name of the pesticide here and in general going forward. Specifically, you used a pesticide (imidacloprid). Therefore, it should be indicated as the effect of imidacloprid.
363-379. The acronyms for Figure 1 are clarified here. It is very late. A legend should be added, as I mentioned previously.
521. "we have."
After thoroughly analyzing the results, I fail to understand why you use the T function to analyze each factor separately and then a mixed model that analyzes everything. Regarding this, wouldn't it be simpler to directly use the mixed model, in which the effects of each factor separately (heat, nutrients, and exposure to imidacloprid) are analyzed, the random factor (which has not been mentioned and could have an influence) and of course, the interactions between effects? With this, you could divide Figure 9 into three figures and visualize the data better (eliminating separate analyses with the T function). It is not clear, for example, if analyzing the W factor, the groups W, WE, WP... within the same W factor have been used, which could introduce a significant bias in the warm vs. no warm analysis. This should be properly justified. Here, only the W group should be compared with the control.
The discussion seems appropriate to me, although in the event of properly reviewing the results, it should be modified accordingly. I find it very appropriate to include a section on limitations and future research.
Despite all the above, I want to congratulate you on your work, which I find very interesting and well done.
Comments on the Quality of English LanguageI have no comments
Round 2
Reviewer 3 Report
Comments and Suggestions for Authors
Dear authors,
After reexamining your manuscript, I believe there is still a recurring issue with the data analysis. I will now proceed with my observations.
Please check minor spelling errors, such as at the end of line 201, "April , " or in line 236, "Brock et al. []," among others.
Figure 2 is excellent.
In lines 245-246, you explicitly mention "with their interactions." It's important to note that the Student's t-test does not involve interactions, as it compares two things, for example, Warm vs. No Warm. Therefore, comparing Warm vs. no Warm in a study with multiple variables (heat, eutrophication, and pesticide) can lead to errors.
Allow me to explain further:
Factor A: Warm (more or less)
Factor B: Eutrophication (more or less)
Factor C: Pesticide (presence or absence)
You state that your analysis focuses on the impact of each factor on zoobenthos abundance. You compare their abundance under different warm levels (Factor A). Then, you compare the abundance under different eutrophication levels (Factor B). Finally, you compare zoobenthos abundance in the presence or absence of a pesticide (Factor C).
However, what if warm affects zoobenthos abundance in a mesocosm where there is a pesticide? Or if eutrophication influences mesocosms with increased temperature? This emphasizes the importance of the linear mixed model (where a random effect, in this case, the month, which also must be clarified for its significance) or conducting tests like ANOVA instead of the Student's t-test, as it may not be the most appropriate. In any case, if a Student's t-test is applied, it would be ideal to compare mesocosm W with the Control only (clearly indicated in the materials and methods). Clearly, ANOVA will also indicate if heat significantly affects zoobenthos abundance, as observed in your study. This consideration also applies to indices.
In lines 252-258, you mention the use of post hoc tests for multiple comparisons. What multiple comparisons should be made between warm and no warm? The same Student's t-test would provide the answer. There seems to be a contradiction in your statement.
I believe this is a data analysis issue that can be addressed, significantly improving the quality of the work. Honestly, the article seems very good to me, but this error (though it probably does not impact your results) should be rectified. In Figure 9, observe how W vs. WP is clearly not equal.
